# Propuesta de Proyecto: Modelado y calibración de transmisión del Virus Respiratorio Sincitial mediante Gramáticas evolutivas y arPSO

**Daniel Parra**
Universidad Complutense de Madrid (UCM)
dparra02@ucm.es

**Marcos Llamazares**
Universitat Politècnica de València (UPV)
marllalo@upvnet.upv.es

**Jose Manuel Velasco**
UCM
mvelascc@ucm.es

**Rafael J. Villanueva**
UPV
rjvillan@imm.upv.es

**Jose I. Hidalgo**
UCM
hidalgo@ucm.es

## 1 Introducción

El Virus Respiratorio Sincitial (VRS) es una de las principales causas de infecciones de las vías respiratorias en niños en todo el mundo, provocando aproximadamente 3 millones de hospitalizaciones y unas 66,000 muertes de menores de cinco años anualmente [1]. Además, representa un problema de salud pública debido a su alta transmisibilidad y marcada estacionalidad [2], lo que genera presión sobre los sistemas sanitarios, especialmente en temporadas de alta incidencia [3]. Aunque actualmente existen estudios epidemiológicos y modelos matemáticos para predecir su propagación, la captura precisa de los picos de infección sigue siendo un desafío [4].

Estos modelos epidemiológicos suelen describir la propagación de enfermedades mediante sistemas de ecuaciones diferenciales que consideran diferentes estados de los individuos (susceptibles, infectados, recuperados, etc.) y parámetros que determinan las tasas de transmisión, recuperación y pérdida de inmunidad. Sin embargo, uno de los aspectos más difíciles de modelar es la variabilidad temporal de la tasa de contagio, que suele representarse con funciones periódicas como cosenos con fase y amplitud ajustables. Aunque estos enfoques han sido ampliamente utilizados, en la práctica, su capacidad para capturar los picos de infecciones observados en los datos empíricos es limitada. Algunos estudios han abordado esta limitación introduciendo múltiples términos sinusoidales o técnicas de inferencia bayesiana para ajustar la tasa de contagio. Sin embargo, estos enfoques incrementan la complejidad del modelo, requieren una mayor cantidad de parámetros a calibrar y, en muchos casos, no consideran adecuadamente la identificabilidad de los parámetros ni la incertidumbre de las estimaciones.

Este proyecto tiene como objetivo desarrollar un modelo de transmisión del VRS en niños menores de un año, empleando un enfoque basado en Gramáticas Evolutivas (GE) [5], una variante de la programación genética [6], para generar funciones de transmisión $\beta(t)$ que representen con mayor precisión la dinámica estacional del virus. Además, se utiliza la Optimización Adaptativa por Enjambre de Partículas con Compartición de Datos Asíncrona y Aleatoria (arPSO)[7] para calibrar los parámetros restantes del modelo epidemiológico, mejorando así la capacidad de predicción del sistema. Esperamos que la combinación de estos enfoques permita una exploración flexible del espacio de búsqueda, asegurando una mejor adaptación a los datos reales sin incrementar innecesariamente la complejidad del modelo.

## 2 Modelo de transmisión

Dependiendo del estado en el que se encuentran los individuos y cómo interactúan, nuestro modelo describe la dinámica de transmisión del VRS a través de distintos estados. En primer lugar, un individuo sano o susceptible (S) puede contagiarse por contacto con un individuo infeccioso (I), tras lo cual pasa al estado latente (L). En esta etapa, el individuo está infectado pero no transmite la

enfermedad; con el tiempo, pasa a ser infeccioso. Un individuo infeccioso es capaz de transmitir la enfermedad y, dependiendo de su evolución, puede recuperarse (R) o, si empeora, puede acabar hospitalizado (H). En el caso de los hospitalizados, tras un periodo de tiempo, deja de ser infeccioso y pasa a estar recuperado. Los individuos recuperados que han superado la infección adquieren una inmunidad temporal y, una vez que esta desaparece, vuelven a ser susceptibles.

En la Figura 1 se muestra la transición de los individuos entre los distintos estados de manera gráfica.

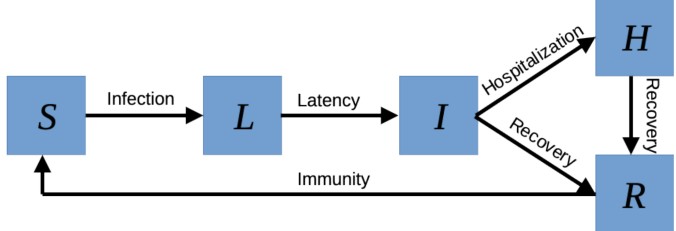

Figure 1: Representación gráfica de la dinámica de transmisión del VRS.

Los parámetros de latencia, hospitalización, recuperación e inmunidad son parámetros constantes que se mueven en unos rangos dependiendo de los individuos. Sin embargo, el contagio sigue un patrón estacional, como se puede ver en la Figura 2, donde se muestran los casos de niños menores de 1 año hospitalizados semanalmente entre 2011 y 2015.

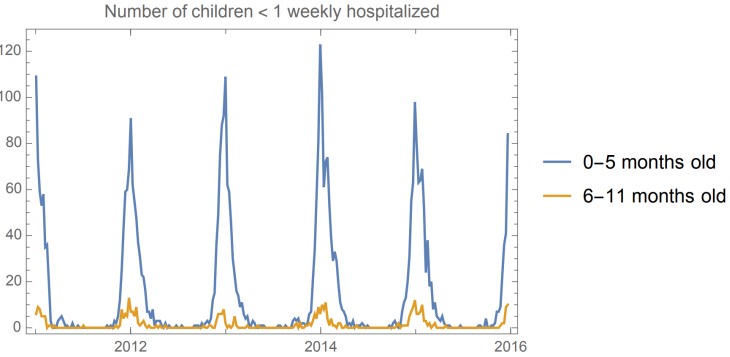

Figure 2: niños menores de 1 año hospitalizados semanalmente entre 2011 y 2015.

La tasa de contagio del VRS se suele modelar mediante una función que involucra cosenos con fase y una amplitud ajustables que permite simular el término estacionar. Sin embargo , estos enfoques pueden no ser suficientes a la hora de capturar los picos de infecciones en los datos. Un enfoque común es el uso de la Ecuación 1:

$$\beta(t) = b_0 + b_1 \cos\left(\frac{2\pi t}{365} + \phi\right) \tag{1}$$

donde $b_0$ es la tasa de contagio base (baseline), $b_1$ es la amplitud y $\phi$ la fase. Este tipo de términos no se han investigado mucho y el caso es que cuando se calibra el modelo, este término estacional no permite alcanzar los picos de infecciosos/hospitalizados, lo cual, desde el punto de Salud Pública y gestión de recursos, es un problema.

## 3   Propuesta

En este proyecto, proponemos el uso de Gramáticas Evolutivas (GE) para calibrar $\beta_i(t)$, permitiendo una mayor flexibilidad en la construcción de la función de transmisión. Aunque en principio sería posible ajustar también otros parámetros del modelo dentro del proceso de GE, consideramos que una estrategia más eficiente es optimizarlos mediante un algoritmo de Optimización por Enjambre de

Partículas Adaptativa (asíncrono-random) (arPSO). Esta aproximación híbrida busca equilibrar la exploración del espacio de búsqueda y la eficiencia computacional en la calibración del modelo.

- Llamaremos $M(\beta_i(t), \alpha_{best}, t)$ el modelo de VRS que depende de la función de transmisión $\beta(t)$ y de una serie de parámetros que vamos a agrupar en el vector $\alpha$, y del tiempo $t$.
- Consideraremos una función de error $E(om, d)$, donde $om$ es la salida del modelo que se va a comparar con alguna medida con los datos $d$.
- Tendremos un algoritmo de optimización mediante gramáticas evolutivas que llamaremos GE con el objetivo de calibrar $\beta(t)$.
- Tendremos el algoritmo de optimización arPSO con el objetivo de calibrar los parámetros contenidos en el vector $\alpha$ .

Partiendo de esta base, a continuación describimos el procedimiento de ajuste empleando GE y arPSO:

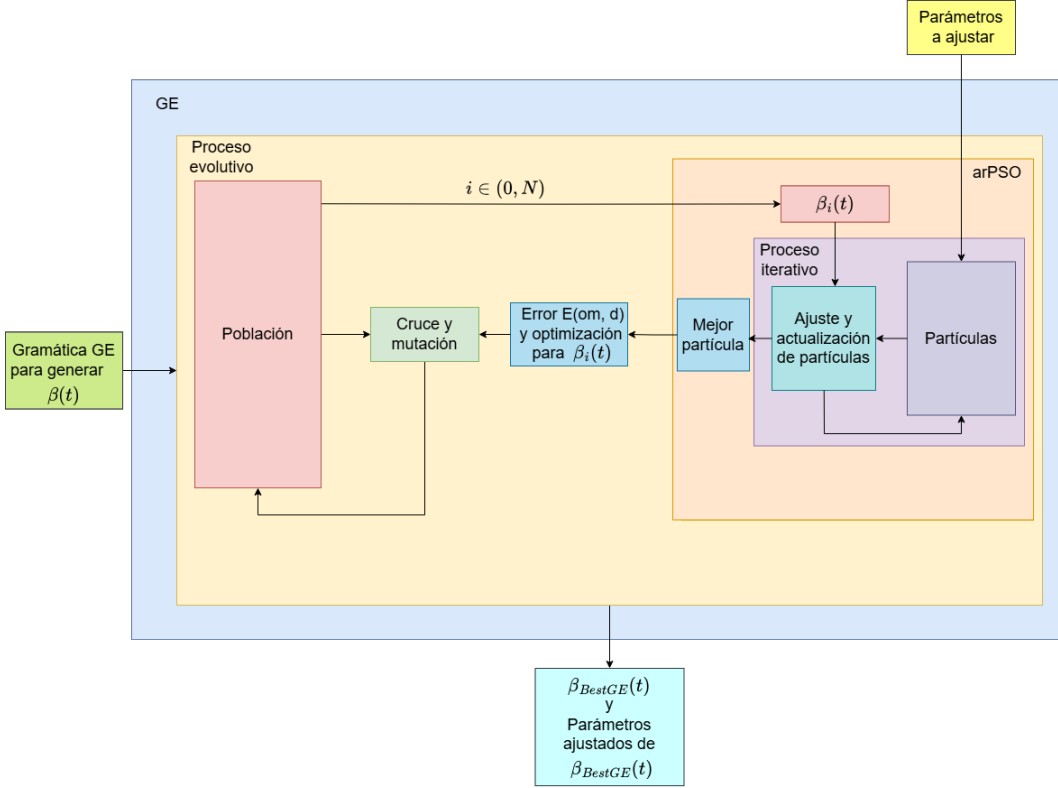

Figure 3: Procedimiento de ajuste de parámetros y obtención de función de transmisión mediante GE y arPSO

1. GE genera N funciones de transmisión $\beta_i(t)$ (individuos).
2. Por cada función se ejecuta una instancia de arPSO con M partículas $\alpha_i$.
   (a) Evaluamos el modelo en cada una de las de partículas $om_i = M(\beta_i(t), \alpha_i, t)$, obteniendo el error de las partículas $e_i = E(om_i, d)$.
   (b) Mediante el proceso de ajuste, se actualiza el mejor resultado y se trata de reducirlo en la medida de lo posible.
   (c) Al cumplir la condición de terminación de arPSO, se devuelve un modelo $om_{Best} = M(\beta_i(t), \alpha_{Best}, t)$ y el error $e_{Best} = E(om_{Best}, d)$ que servirá como error para la $\beta_i(t)$ de GE.
3. Se genera la nueva población de GE mediante cruce y mutación, tomando como fitness el error obtenido por el arPSO para los diferentes individuos.

4. Volvemos al paso 2 y repetimos hasta cumplir la condición de finalización de GE.

Al finalizar el proceso descrito, representado en la Figura 3, obtendremos una función de transmisión $\beta(t)$ y su configuración óptima de parámetros con los que construimos el modelo $om$.

## 4  Agradecimientos

Marcos Llamazares López ha sido beneficiario de una beca de doctorado por parte del Programa de Ayudas de Investigación y Desarrollo (PAID), Universitat Politècnica de València (UPV). Este trabajo ha contado con el apoyo del Ministerio de Innovación, Ciencia y Universidad de España (PID2021-125549OB-I00) y de los Fondos Next Generation de la UE (PDC2022-133429-I00).

## 5  Requisitos

Para la ejecución del proyecto, se requieren los siguientes elementos clave:

- **Desarrollo del entorno de simulación**: Es fundamental implementar una plataforma que integre Gramáticas Evolutivas (GE) y Optimización por Enjambre de Partículas Adaptativa (arPSO), permitiendo la calibración eficiente del modelo epidemiológico.
- **Validación del enfoque propuesto**: Se debe evaluar si la combinación de GE y arPSO produce soluciones precisas y útiles en la estimación de la función de transmisión $\beta(t)$ y los parámetros del modelo.
- **Desarrollo de una gramática adecuada**: Es crucial definir una gramática que permita generar expresiones funcionales relevantes, incorporando operadores matemáticos como constantes, términos trigonométricos (senos y cosenos) y exponenciales, asegurando la capacidad de representar patrones epidemiológicos complejos.
- **Análisis del espacio de búsqueda**: Se debe garantizar que la estructura modular del modelo no introduzca restricciones que perjudiquen la exploración y optimización conjunta de los parámetros y la función de transmisión.
- **Estudio de la convergencia**: Es necesario analizar la estabilidad y eficiencia conjunta de ambos algoritmos, verificando que converjan hacia soluciones óptimas en un tiempo computacional razonable.

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
