# OpenReview forum: "Modelado y calibración de transmisión del virus respiratorio sincitial mediante Gramáticas evolutivas y arPSO"
_MAEB/2025/Projects_Track — MAEB 2025 Proyectos_

### Official Review · Reviewer_q31X · 2025-03-17
**Hybrid approach to deal with a real-word health problem**

**Rating:** 5
**Confidence:** 4

**Review:**

The project proposal is focused on tackling a public health problem, so its interest is very high. The combined use of evolutionary grammars with a variant of the PSO algorithm makes the thematic fit with the topics of MAEB.

The only aspect that remains unclear is why the arPSO algorithm has been specifically chosen instead of a standard PSO or some other metaheuristic oriented to continuous optimisation such as differential evolution (DE). The only suggestion is to consider other possible algorithms in case the results obtained by using arPSO are not satisfactory.

---

### Official Review · Reviewer_D51P · 2025-03-17
**Propuesta interesante con algunos puntos a mejorar**

**Rating:** 4
**Confidence:** 4

**Review:**

La propuesta de proyecto es interesante, pero sería deseable la mejora de algunos aspectos:
- Llama la atención que no se describe el conjunto de datos a utlizar o su naturaleza y la forma de obtención.
- Debería incluirse le fuente bibliográfica de la Ecuación 1.
- Errores tipográficos: hay algunos errores tipográficos a corregir. Por ejemplo:
   - 60,000  -> 60.000
   -  Sin embargo , -> Sin embargo,
  Revisar el texto detenidamente.

---

### Official Review · Reviewer_eJha · 2025-03-17
**Modelado y calibración de transmisión del virus respiratorio sincitial mediante Gramáticas evolutivas y arPSO**

**Rating:** 4
**Confidence:** 4

**Review:**

Este proyecto presenta un enfoque innovador que combina Grammatical Evolution (GE) con la optimización de parámetros mediante asynchronous random Particle Swarm Optimization (arPSO) para generar la función de transmisión y perfeccionar automáticamente el modelo del VRS. Al integrar computación evolutiva con optimización heurística, el método propuesto busca desarrollar modelos matemáticos interpretables que equilibren precisión y capacidad de generalización. Se espera que los resultados de esta investigación mejoren la predicción epidemiológica, especialmente en la identificación de picos de incidencia estacional, facilitando así la toma de decisiones institucionales y fortaleciendo las estrategias de vigilancia e intervención contra el VRS.

El proyecto resulta interesante y factible en los términos definidos mediante sus fases de desarrollo y alcance. No obstante, se recomienda la formulación clara de una hipótesis de partida, a lo sumo de una serie de preguntas de investigación, con sus indicadores de obtención, para enriquecer su definición. También se recomienda definir de manera clara la metodología de desarrollo, pruebas y experimentación.

Por último, un aspecto clave en este tipo de proyectos basados en datos de naturaleza médica, consiste en explicar de manera detallada la estrategia de datos a seguir, es decir, si se disponen de datos reales o realistas, definir en su caso las fuentes, si son de naturaleza sensible, su tratamiento, si se utilizarán en conjunto con datos sintéticos o de benchmarking, etc.

---

### Decision · Program_Chairs · 2025-03-20

Accept